# The Mechanism of Androgen Actions in PCOS Etiology

**DOI:** 10.3390/medsci7090089

**Published:** 2019-08-28

**Authors:** Valentina Rodriguez Paris, Michael J. Bertoldo

**Affiliations:** 1Fertility and Research Centre, School of Women’s and Children’s Health, University of New South Wales Sydney, NSW 2052, Australia; 2School of Medical Sciences, University of New South Wales Sydney, NSW 2052, Australia

**Keywords:** PCOS, androgens, fertility

## Abstract

Polycystic ovary syndrome (PCOS) is the most common endocrine condition in reproductive-age women. By comprising reproductive, endocrine, metabolic and psychological features—the cause of PCOS is still unknown. Consequently, there is no cure, and management is persistently suboptimal as it depends on the ad hoc management of symptoms only. Recently it has been revealed that androgens have an important role in regulating female fertility. Androgen actions are facilitated via the androgen receptor (AR) and transgenic *Ar* knockout mouse models have established that AR-mediated androgen actions have a part in regulating female fertility and ovarian function. Considerable evidence from human and animal studies currently reinforces the hypothesis that androgens in excess, working via the AR, play a key role in the origins of polycystic ovary syndrome (PCOS). Identifying and confirming the locations of AR-mediated actions and the molecular mechanisms involved in the development of PCOS is critical to provide the knowledge required for the future development of innovative, mechanism-based interventions for the treatment of PCOS. This review summarises fundamental scientific discoveries that have improved our knowledge of androgen actions in PCOS etiology and how this may form the future development of effective methods to reduce symptoms in patients with PCOS.

## 1. Introduction

Polycystic ovary syndrome (PCOS) is a heterogeneous condition, which affects approximately 6% to 20% of women of reproductive age. It is the most common endocrine condition in women of this age group [1,2,3]. Although there have been numerous diagnostic criteria for PCOS from the Androgen Excess and PCOS Society (AE-PCOS), Rotterdam, and the National Institutes of Health (NIH) (NIH criteria), an international evidence-based guideline for the assessment and management of PCOS was released in late 2018 which endorses the use of the Rotterdam diagnostic criteria [4]. For a woman to be diagnosed with PCOS, she must exhibit two out of the following three PCOS features: clinical and/or biochemical androgen excess, oligo-ovulation or anovulation, and polycystic ovarian morphology (PCOM) on ultrasound [3,5]. Nevertheless, PCOS can be accompanied by a much wider range of co-morbidities, as patients with PCOS can also display reproductive, endocrine, metabolic and psychological features [5,6]. Disturbed hormonal and reproductive features include luteinizing hormone (LH) excess, hyperandrogenism, ovulatory perturbations, aberrant follicular development, diminished fertility, and an increased risk of miscarriage [2]. Moreover, if pregnancy is attained, women with PCOS have a considerably greater risk of pregnancy difficulties, including gestational diabetes, hypertensive disorders and premature delivery [7]. PCOS also has a significant metabolic component as it is linked with obesity, metabolic syndrome, hyperinsulinemia, insulin resistance, hepatic steatosis and dyslipidemia, which amplify the risk of type-2 diabetes and cardiovascular disease [2,8,9]. Furthermore, the long-term risk of PCOS-related defects in offspring from women with PCOS is an area of increasing interest, as existing data implies that maternal PCOS is linked with a susceptibility to unfavourable PCOS-associated health consequences in their children [10].

Notwithstanding the pervasiveness of PCOS and its noteworthy health impact, a cure for PCOS does not exist, and current strategies for its management are insufficient, as they depend on ad hoc and imperfect treatment of symptoms. Given no drug has ever been approved specifically for the PCOS condition [3], the majority of drugs used to treat PCOS symptoms are administered in an off-label fashion. Currently, the genesis of PCOS continues to evade us; therefore, mechanism-based interventions persist elusively. A significant need exists for continuing research to characterise the etiology of PCOS, and hence deliver the required knowledge for the advancement of mechanism-based directed interventions for this condition.

## 2. The Relationship between Hyperandrogenism and the Pathophysiology of PCOS

Hyperandrogenism represents a chief attribute of PCOS as elevated androgen levels are the most constant feature, with the majority (−60%) of patients exhibiting hyperandrogenism (Rotterdam definition) [11]. Women with hyperandrogenic PCOS present with elevated levels of various androgens, including testosterone (T) and the pro-androgens androstenedione (A_4_) and dehydroepiandrosterone sulfate (DHEAS), as well as the enzyme required to convert pro-androgens to bioactive androgens, 3β-hydroxysteroid dehydrogenase (3β-HSD) in serum [12,13]. Excess androgens can be induced by insulin resistance and hyperinsulinemia, as they cause a reduction in sex hormone binding globulin levels, which lead to a subsequent increase in free androgens and unfavourable metabolic profiles [14,15]. The ovarian PCOS morphological traits of enlarged, multi-cystic ovaries and theca interstitial hyperplasia are reported in women who are subjected to high levels of androgens as a result of endogenous adrenal androgen hypersecretion in congenital adrenal hyperplasia [16], or exogenous testosterone treatment in female-to-male transsexuals [17,18]. Additionally, cultured human theca interna cells removed from PCOS ovaries exhibit higher androgen secretion that continues during long-term culture [19]. These observations corroborate a role for androgens in the acquisition of the PCOS ovarian features. 

## 3. Clinical Targeting of Androgen Excess—Potential for Mitigating Against PCOS

Androgen receptor antagonists have proven useful in the treatment of PCOS phenotypes. Through its actions on the hypothalamus, pituitary and ovarian steroidogenesis, the use of the oral contraceptive pill has the overarching effect of reducing hyperandrogenism [20]. These effects make the oral contraceptive pill an effective pharmacological intervention for the treatment of menstrual irregularity, hirsutism, acne and androgenic alopecia associated with PCOS [21,22,23,24]. In PCOS patients, third generation combined oral contraceptive pills that comprise antiandrogenic compounds, have demonstrated a beneficial effect on the metabolic phenotypes of PCOS, with patients displaying enhanced lipid and adipokine profiles [22]. However, oral contraceptive pills are suitable only for those females who are not attempting to conceive. For those patients desiring to conceive, the administration of AR blockers such as spironolactone, cyproterone acetate and flutamide, or the 5 alpha-reductase inhibitor finasteride have shown similar efficacy for ameliorating the adverse effects of PCOS when treating PCOS patients [3,21].

Flutamide, a competitive antagonist of the AR, is most widely used and has been reported to have a favourable effect in women with PCOS as it decreases hirsutism and acne [25,26,27]. PCOS patients under flutamide treatment also experienced improved menstrual cycle regularity and ovulation [28,29]. Additionally, treatment of both obese and lean PCOS women with flutamide revealed that independent of weight changes, flutamide improved the lipid profile of women with PCOS, with a significant decrease in total cholesterol, low-density lipoproteins (LDL) and triglycerides [30]. The different effects observed between treatments are summarised in Figure 1. However, interventions that reduce levels of excess androgens may still not be effective in patients desiring children as it appears hyperandrogenism may impart adverse legacies on fertility, even after follicles and oocytes have been removed from the hyperandrogenic environment [31].

Alternative anti-androgens include spironolactone and cyproterone acetate, steroidal AR blockers that compete with T and DHT for binding the AR. Both of these AR blockers are observed to be effective in significantly decreasing levels of hirsutism and acne in PCOS patients [3,21]. Moreover, spironolactone therapy in one study was reported to positively improve metabolic traits in women with PCOS [32]. Another treatment used to appease hyperandrogenic symptoms of PCOS is finasteride, a 5-alpha reductase inhibitor that prevents the conversion of T to DHT (a more potent androgen), and is effective for the management of hirsutism in PCOS patients [33,34].

Taken together, results from the use of anti-androgenic drugs in PCOS patients either alone or in combination have demonstrated that targeted inhibition of hyperandrogenism, and therefore androgenic actions, has a beneficial effect with improvements observed in numerous PCOS traits. Since the AR facilitates the actions of androgens, these observations provide support for an association between hyperandrogenism and the development of an extensive array of PCOS traits. Nevertheless, these treatments are not restorative, and such pharmacological interventions simply provide an opportunity for PCOS women to alleviate PCOS symptoms. However, evidence suggests that antiandrogens have unacceptable hepatotoxicity [21], which offsets their benefits for use in non-lethal disorders, such as PCOS. Therefore, although universal androgen blockade is a rational strategy for treating PCOS, a more targeted pharmacological approach is required. However, this necessitates a profound understanding of the biological mechanisms supporting its development.

## 4. Development of Pre-Clinical Animal Models of PCOS

A powerful method to study PCOS is through animal models that mimic the features present in the human condition of PCOS. Animal research is necessary as research in humans is hampered due to the inability to perform fully controlled human studies that have considerable ethical and logistical restrictions. Studies using animal models provide insights into basic biological mechanisms stimulating the development of PCOS, and thus provide the opportunity to locate novel targets for the treatment of PCOS that surpass current medical practice of symptom-based treatments. While animal models of PCOS have been developed using different approaches including treatment with androgens, estrogens, antiprogestins and genetic manipulation [35,36], the most relevant information on the genesis of PCOS has resulted from PCOS animal model studies developed using androgen excess [35,37,38].

Hyperandrogenism has been postulated to play a key role in the origins of PCOS, as PCOS animal models created through the induction of hyperandrogenism consistently produce animals that present a broad range of reproductive, endocrine and metabolic features of PCOS [39]. These observations are appreciated in a variety of mammalian species, including rodents, sheep and rhesus monkeys, in which exposure to high levels of androgens during prenatal or early postnatal life has successfully induced a breadth of PCOS features.

PCOS is most closely recapitulated in animal models with either prenatal exposure to testosterone or DHT, or early postnatal exposure to DHEA or DHT in rodents, and in sheep and nonhuman primates by excess prenatal exposure to testosterone (Figure 2). These animal models effectively exhibit a PCOS-like phenotype, however, variations in the appearance of PCOS features is observed depending on the time of androgen excess exposure, suggesting that there are developmental windows of androgen exposure that are key in the pathogenesis of PCOS. Rodents, sheep and primates exposed prenatally to excess levels of androgens have been reported to exhibit most of the key endocrine, reproductive, and metabolic traits of PCOS found in women [35,36,37,38,40]. These studies support a foetal origin for the genesis of PCOS and postulate that normal programming during gestation is altered by excess levels of androgens in utero, leading to aberrant reproductive, endocrine and metabolic function in women [41,42]. Prenatally androgenized rodent, sheep and nonhuman primate models of PCOS all exhibit the clinical features of hyperandrogenism, LH hypersecretion and the development of the classic polycystic ovarian morphology of numerous arrested antral follicles and ovulatory dysfunction [36].

Metabolic features of PCOS are also present in animal models of PCOS with obesity and aberrant adipose function observed in several androgen-induced PCOS animal models. Androgenized female mice display increased fat mass, enlarged adipocyte cells accompanied by decreased adiponectin levels inferring adipose tissue dysfunction [43,44,45,46,47], whereas the rhesus monkey PCOS model exhibits hindered preadipocyte differentiation [48]. Moreover, insulin resistance and glucose intolerance have emerged as a result to excess androgen levels in rodent, sheep and primate PCOS animal models, once more mimicking the clinical PCOS traits [36,37,38].

Additional evidence promoting androgens as significant drivers in the etiology of PCOS come from studies reporting that treatment with AR antagonists on PCOS animal models prevented or reversed the manifestation of some PCOS traits. For example, the prenatally androgenized PCOS sheep model with ovulatory dysfunction, displayed restoration of LH surges necessary for ovulation when co-treated with the AR antagonist flutamide [49]. In a mouse model of PCOS, co-treatment with flutamide restored estrous cycling, reduced the number of cyst-like follicles in the ovaries and reduced body weight and adipocyte size [50]. In the brain, the use of flutamide has also been observed to amend changes in GABAergic drive to GnRH neurons in a PCOS mouse model [51], as well as changes observed in agouti-related peptide (AgRP) neurons in a PCOS sheep model [52]. Together, these intervention studies demonstrate that AR mediated actions play a significant role in the development of PCOS features in experimental models.

## 5. Insights into the Origin of PCOS from Studies in Pre-Clinical PCOS Animal Models

Recent research has focused on identifying the site(s) of androgen action involved in the development of PCOS. A number of recent studies have incorporated the use of androgen receptor knockout (ARKO) mouse models as a means to explain the direct role of androgens in PCOS development, since androgen actions are directly mediated via the AR. It was observed in global ARKO female mice exposed to androgen excess that PCOS could not be induced [45,53], confirming that a functional AR is needed to develop a PCOS animal model. Moreover, androgen excess has been suggested to act via the AR at different locations in the body, such as the hypothalamus, ovary, adipocyte cells and/or skeletal muscle, and give rise to PCOS.

To explore the location of these sites, recent studies have used PCOS mouse models in combination with global and cell-specific ARKO mouse models to better understand the mechanisms behind the androgen-induced PCOS environment. To locate the main sites of AR actions, PCOS has been induced in mice with a non-functional AR in either ovarian granulosa cells, theca cells or in the brain. Female mice with a loss of AR function only in the granulosa cells displayed the majority of PCOS characteristics and were only safeguarding against increased granulosa cell degeneration in antral follicles [45]. Inactivation of AR signalling in ovarian theca cells was observed to be able to only partially prevent the development of acyclicity, ovulatory dysfunction and infertility in an androgen induced PCOS mouse model [45,54]. These findings indicate that the ovary is not the primary pathophysiological site for the development of PCOS. However, silencing of AR actions in the brain inhibited the appearance of most reproductive and metabolic traits of PCOS, pinpointing the brain as a leading site in the pathophysiology for developing PCOS [45,55]. Further evidence ruling out ovaries as the major pathological site of androgen actions in PCOS development was revealed in elegant experiments using ovarian transplants. PCOS was observed to still develop in animals when transplanting ARKO ovaries (with a non-functional AR) into ovariectomized hyperandrogenic wild-type mice (with a functional AR)—animals which had a functional AR in all tissues except for in the ovary [45]. In contrast, in hyperandrogenized and ovariectomized global ARKO mice that received control ovaries containing a functional AR, PCOS did not develop [45]. These observations further corroborate that extra-ovarian mechanism are the main drivers in the development of PCOS. Moreover, these findings indicate that while ovarian AR signalling may be involved in the development of reproductive features observed in PCOS, neuroendocrine androgen-driven mechanisms in the brain are the key mediators in the developmental origins of PCOS.

## 6. Translation of Basic Research in PCOS for the Development of Androgen-Targeted Interventions

There is significant evidence advocating for androgen excess mediated actions through the AR in the origin of PCOS. However, current generations of anti-androgens are systemic and, as mentioned earlier, lead to unacceptable liver toxicity, which disqualifies them for use in non-lethal chronic disorders such as PCOS. Therefore, the focus of current research has been directed at identifying ways to specifically target and suppress hyperandrogenic effects in women with PCOS.

### 6.1. Neuroendocrine Pathways

A potentially effective strategy is to target AR-signalling in neuroendocrine pathways, as specific loss of AR actions in the brain prevented hyperandrogenized mice from developing PCOS traits, identifying the brain as a main site involved in experimental PCOS [45]. Women with PCOS, often display an increase in LH to FSH ratio and LH pulse frequency [2], which is also observed in rat [56], mouse [40,57] and sheep [58] PCOS models. Although GnRH neurons regulate LH and FSH secretion, they do not express AR [59]. AR are expressed in the upstream neuronal network, the kisspeptin-neurokinin B (NKB)-Dynorphin “KNDy” system within the arcuate nucleus, which is involved in GnRH secretion [60,61,62]. Studies have shown that AR-mediated signalling participates in the regulation of the KNDy system [63] and increased kisspeptin is observed in some PCOS patient cohorts [64,65,66]. Furthermore, its expression and circuitry has been reported to be altered in hyperandrogenized rodent and sheep PCOS animal models [46,67,68,69]. These results denote the KNDy system as a potential therapeutic target to lessen AR-driven neuroendocrine actions in women with PCOS. Yet of note, not all population studies of women with PCOS have reported changed kisspeptin levels; however, this could be due to variations in PCOS phenotype. Nonetheless, a recent clinical study treating PCOS patients with a neurokin-3 (NK3) receptor antagonist reduced LH and T concentrations plus LH pulse frequency [70], favouring the KNDy system as a site of target.

### 6.2. Metabolic Pathways

Specific loss of AR signalling in the brain also protects hyperandrogenized PCOS mice from developing metabolic PCOS traits, suggesting that metabolic dysfunction in PCOS patients may also be mediated via central mechanisms regulated by the AR [45]. There is even evidence proposing that an androgen-brain-adipocyte axis might be involved in the development of metabolic dysfunction observed in PCOS. In a mouse model of androgen excess, leptin, a hormone predominantly made in adipocytes and involved in energy homeostasis, has reduced homeostatic capabilities as it failed to increase expression of uncoupling protein-1 in brown adipose tissue (BAT) [43]. Furthermore, leptin is known to target the proopiomelanocortin (POMC) and neuropeptide Y/Agouti-related peptide (NPY/AgRP) neurons [71]. In a PCOS mouse model, POMC mRNA and fiber projections were reduced [43]. In contrast, in a sheep model of PCOS, androgen excess led to increased NPY/AgRP cell number and fiber projections [52]. However, flutamide treatment blocks the NPY/AgRP neuron changes in the sheep PCOS model [52]. Collectively, these studies support the premise that central AR driven mechanisms are involved in the development of metabolic traits in PCOS and should be further investigated.

Androgen receptor-driven actions occurring in adipose tissue are also of interest for the development of new strategies to treat PCOS. Studies have shown that the observed increase in intra-abdominal fat mass in women with PCOS, is positively correlated with circulating androgen levels [72]. Changes in adipocyte morphology and/or function are also replicated in hyperandrogenized rodent, sheep and primate models of PCOS [45,48,73]. Interestingly, modifications in adipocyte morphology are observed before the onset of insulin insensitivity in the sheep model, therefore, aberrant adipocyte function may be present before the start of metabolic dysfunction [73]. This information stresses the importance of conducting research that identifies the mechanisms underpinning these alterations, as it may potentially lead to treatment of PCOS from an earlier stage and thus prevent its progression.

Another potential androgen excess-AR-driven mechanism leading to aberrant adipose tissue function in PCOS could be the inability of adipocytes to generate adequate levels of the adipokines. Adiponectin, an adipokine involved in glucose and lipid metabolism, has been observed to be lower in women with PCOS [74], and several PCOS mouse models [45,75,76]. However, this decrease was reversed in a study where brown adipose tissue (BAT) from healthy control rats was transplanted into hyperandrogenic PCOS female rats, resulting in enhanced BAT activity, increased serum adiponectin levels and rescue of several PCOS traits such as irregular cycles and insulin resistance [77]. Additionally, exogenous adiponectin administration reiterated the favourable effects that came from BAT transplantation [77]. This result was also replicated in another study utilising a hyperandrogenized PCOS mouse model [75]. The importance and potential therapeutic approach by modulation of adipokines is further drawn in transgenic studies where overexpression of adiponectin protected hyperandrogenized mice from developing metabolic PCOS traits, while lack of adiponectin lead to amplified or comparable features to those observed in the androgen-induced PCOS mouse model [76]. However, investigations focusing on consequential mechanisms are required.

## 7. Future Perspectives

New innovative approaches aimed at reducing overall androgen excess have been tested in pre-clinical research and some clinical trials. One compound tested is dietary medium-chain fatty acid (decanoic acid), which reduces androgen biosynthesis in vitro by modulating the actions of the steroidogenic enzyme 3β-HSD leading to androgen reduction [78]. A study using decanoic acid as a treatment in a rat model of PCOS reported restoration of cyclicity and reduction of T and fasting insulin levels [78]. Another possible novel therapeutic is resveratrol, a polyphenol whose action is to inhibit androgen production by lowering CYP17 and CYP21 protein expression and activity [79]. Moreover, resveratrol has been observed to improve some ovarian features in a rat model of PCOS [80]. The beneficial effects of resveratrol, a sirtuin (SIRT) 1 activator, have also been reported in a clinical trial conducted in women with PCOS where resveratrol led to a decrease in total T, fasting insulin levels, and also increased the insulin sensitivity index in these women [81]. The Sirtuin group of proteins possesses important roles in the regulation of both metabolic and reproductive functions; therefore, further work is required to fully dissect the roles of the SIRTs in PCOS for the development of innovative interventions.

## 8. Conclusions

There is significant evidence from human and animal studies demonstrating that excess androgens through the AR play a key role in the origin of PCOS (Figure 3). Research using hyperandrogenic animal models of PCOS have advanced our understanding of the genesis of this condition, and have provided insights into potential new treatments for PCOS. With continued research, the combination of clinical observations and basic science will give way to defining the specific androgenic mechanisms governing female reproductive function and how to develop target specific treatments for women with PCOS.

## Figures and Tables

**Figure 1 medsci-07-00089-f001:**
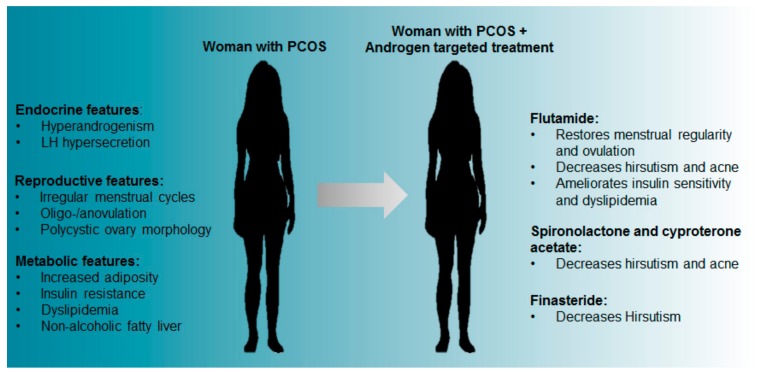
PCOS features and the reported effects of androgen blocking treatments in women with PCOS.

**Figure 2 medsci-07-00089-f002:**
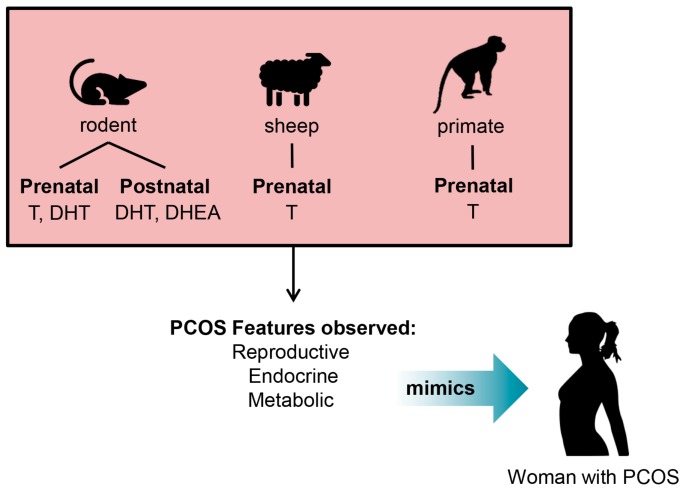
Proposed optimal period for exposure of excess androgens to induce the PCOS phenotype in animal models. Diagram of rodent, sheep and primate PCOS animal models and their prenatal or postnatal period of exposure to androgens to give rise to the PCOS phenotype observed in humans. T = Testosterone DHT = Dihydrotestosterone DHEA = Dehydroepiandrosterone.

**Figure 3 medsci-07-00089-f003:**
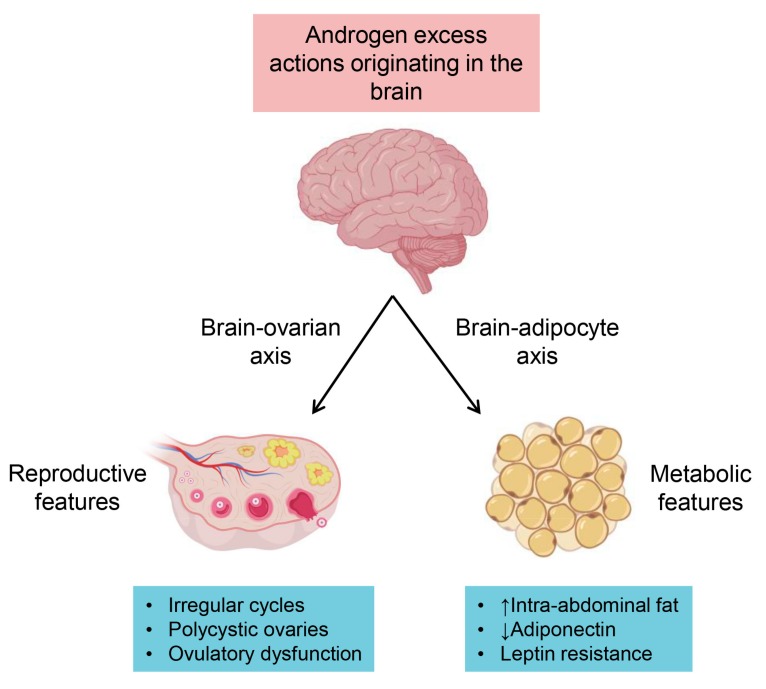
The most recent understanding of androgen receptor mechanisms proposed to be involved in PCOS trait development. Here we illustrate the two proposed axes involved in the development of reproductive and metabolic features of PCOS resulting from androgen excess and the consequential PCOS traits developed.

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
