# Peer review of "The Mechanism of Androgen Actions in PCOS Etiology"

_medsci, 2019, doi:10.3390/medsci7090089_

Round 1

Reviewer 1 Report

In this review, the author summarizes fundamental scientific discoveries that have improved our knowledge of 24 androgen actions in PCOS etiology and how this may form the future development of effective 25 methods to reduce symptoms in patients with PCOS. However, some issues need to improve in this manuscript.

1.          In the section of “Development of pre-clinical animal models of PCOS” the DHEA-induced PCOS also need to include in this section.

2.          The totally summary figure need to include in this manuscript.

Author Response

REVIEWER 1:

In the section of “Development of pre-clinical animal models of PCOS” the DHEA-induced PCOS also need to include in this section

We have included DHEA as suggested in the script and have consequently also updated figure 2.

Lines 140-142 now read as:

“PCOS is most closely recapitulated in animal models with either prenatal exposure to testosterone or DHT, or early postnatal exposure to DHEA or DHT in rodents…..”

The totally summary figure need to include in this manuscript

We agree with the reviewer and have added figure 3 referenced in line 300 as follows:

There is significant evidence from human and animal studies demonstrating that, excess androgens through the AR, play a key role in the origin of PCOS (Figure 3).

Reviewer 2 Report

The authors provide an updated review on the studies addressing the role of androgens and androgen receptor in the pathogenesis of polycystic ovary syndrome. 

·      Minor comments

1.     Abstract, line 12: “Comprising reproductive, endocrine, metabolic …” Please revise punctuation.

2.     While in Figure 1 the authors mention the most used antiandrogens in polycystic ovary syndrome women, in the text they address only flutamide. Please clarify.

3.     Line 278. “steroidogenic enzyme HSD3B2”. Please be consistent with line 67 (in which it is written as 3β-HSD). 

4.     The authors failed to mention that insulin-resistance and hyperinsulinemia lead to a reduction in sex hormone binding globulin levels, with subsequent increase of free androgens levels. Particularly, polymorphisms of the insulin receptor substrate-1, such as Gly972Arg, are associated with increased androgens levels and unfavorable metabolic profile (see Horm Metab Res. 2010 Jul;42(8):575-84; J Endocrinol Invest. 2017 Apr;40(4):367-376; J Clin Transl Endocrinol. 2018 May 24;13:1-8).

Author Response

REVIEWER 2:

Abstract, line 12: “Comprising reproductive, endocrine, metabolic …” Please revise punctuation

We agree with the reviewer and have modified punctuation as follows:

Page 1, lines 12-13.

Comprising, reproductive, endocrine, metabolic and psychological features, the cause of PCOS is unknown

While in Figure 1 the authors mention the most used antiandrogens in polycystic ovary syndrome women, in the text they address only flutamide. Please clarify.

We agree it is important to address in the manuscript text the antiandrogens mentioned in Figure 1, and we have incorporated the following information in lines 99-105: 

Alternative anti-androgens include spironolactone and cyproterone acetate, steroidal AR blockers that compete with T and DHT for binding the AR. Both of these AR blockers are observed to be effective in significantly decreasing levels of hirsutism and acne in PCOS patients [3, 22]. Moreover, spironolactone therapy in one study was reported to positively improve metabolic traits in women with PCOS [33]. Another treatment to appease hyperandrogenic symptoms of PCOS is finasteride, a 5-alpha reductase inhibitor that prevents the conversion of T to DHT (a more potent androgen), and is effective for the management of hirsutism in PCOS patients [34, 35].

Line 278. “steroidogenic enzyme HSD3B2”. Please be consistent with line 67 (in which it is written as 3β-HSD).

We agree with the reviewer and have modified punctuation as follows:

Page 8, line 278 (now line 286).

One compound tested is dietary medium-chain fatty acid (decanoic acid) which reduces androgen biosynthesis in vitro by modulating the actions of the steroidogenic enzyme 3β-HSD HSD3B2, leading to androgen reduction [81].

The authors failed to mention that insulin-resistance and hyperinsulinemia lead to a reduction in sex hormone binding globulin levels, with subsequent increase of free androgens levels. Particularly, polymorphisms of the insulin receptor substrate-1, such as Gly972Arg, are associated with increased androgens levels and unfavorable metabolic profile (see Horm Metab Res. 2010 Jul;42(8):575-84;J Endocrinol Invest. 2017 Apr;40(4):367-376; J Clin Transl Endocrinol. 2018 May 24;13:1-8).

We have included the references as suggested on Lines 67-70 as follows:

Excess androgens can be induced by insulin resistance and hyperinsulinemia, as they cause a reduction in sex hormone binding globulin levels, which lead to a subsequent increase in free androgens and unfavourable metabolic profiles [15, 16].

Thank you very much for your time and trouble in considering this manuscript.

Yours sincerely,

Dr Michael Bertoldo

Lecturer